# Unsupervised Nonlinear Hyperspectral Unmixing with Reduced Spectral Variability via Superpixel-Based Fisher Transformation

Zhangqiang Yin 🆔 and Bin Yang *🆔

School of Computer Science and Technology, Donghua University, Shanghai 201620, China;
2222736@mail.dhu.edu.cn
* Correspondence: yangb19@dhu.edu.cn

**Abstract:** In hyperspectral unmixing, dealing with nonlinear mixing effects and spectral variability (SV) is a significant challenge. Traditional linear unmixing can be seriously deteriorated by the coupled residuals of nonlinearity and SV in remote sensing scenarios. For the simplification of calculation, current unmixing studies usually separate the consideration of nonlinearity and SV. As a result, errors individually caused by the nonlinearity or SV still persist, potentially leading to overfitting and the decreased accuracy of estimated endmembers and abundances. In this paper, a novel unsupervised nonlinear unmixing method accounting for SV is proposed. First, an improved Fisher transformation scheme is constructed by combining an abundance-driven dynamic classification strategy with superpixel segmentation. It can enlarge the differences between different types of pixels and reduce the differences between pixels corresponding to the same class, thereby reducing the influence of SV. Besides, spectral similarity can be well maintained in local homogeneous regions. Second, the polynomial postnonlinear model is employed to represent observed pixels and explain nonlinear components. Regularized by a Fisher transformation operator and abundances' spatial smoothness, data reconstruction errors in the original spectral space and the transformed space are weighed to derive the unmixing problem. Finally, this problem is solved by a dimensional division-based particle swarm optimization algorithm to produce accurate unmixing results. Extensive experiments on synthetic and real hyperspectral remote sensing data demonstrate the superiority of the proposed method in comparison with state-of-the-art approaches.

**Keywords:** hyperspectral imagery; nonlinear spectral unmixing; spectral variability; Fisher transformation; superpixel segmentation



## 1. Introduction

Containing hundreds of continuous and narrow spectral bands, hyperspectral imagery (HSI) has superior discriminatory capability in the identification of diverse materials compared to multispectral images. HSI has been widely investigated and successfully applied in various fields, e.g., image fusion [1], image recovery [2], super-resolution [3], classification [4], and target detection [5]. Nevertheless, due to its inherent limitation of poor spatial resolution, more than one material inevitably exists in a pixel, which reduces the precision of traditional pixel-level remote sensing tasks. Hyperspectral unmixing (HU) is adopted to address this issue by decomposing mixed pixels into a set of pure components (i.e., endmembers) and their corresponding abundance fractions [6].

In an ideal condition, observed pixels can be simply approximated by the well-known linear mixture model (LMM) assuming that a pixel is the linear combination of endmembers based on their corresponding abundances. The LMM has gained popularity in unmixing due to its physical interpretability and straightforward mathematical formulation [7]. Typically, many studies have tried to jointly exploit spectral and spatial information in the framework of nonnegative matrix factorization (NMF) by using the total variation

(TV) [8], graph theory [9], and similarity weighting [10,11], etc., which are promising for the improvement of linear unmixing. However, linear unmixing results may be poor in many real scenarios (e.g., complex urban areas or intimately mixed minerals) where nonlinear mixing effects and spectral variability (SV) are unneglectable.

To explain the nonlinear mixing effects, nonlinear mixture models (NMMs) have been developed. For example, Heylen and Gader [12] incorporated a simplified Hapke model into the LMM to explain light's multiple scatterings in intimate mixtures. Bilinear mixture models (BMMs) such as the generalized bilinear model (GBM) [13] and the polynomial postnonlinear model (PPNM) [14] extend the LMM by considering second-order scatterings. The $p$-order polynomial model [15] and the multilinear mixing (MLM) model [16] account for high-order interactions among endmembers. Recently, considerable endeavors have been made to enhance NMM-based unmixing, including the analysis of band-wise nonlinearity [17], kernel-transformed BMMs [18], and robustness to complex noise [19], etc. In addition, the use of deep learning (DL) techniques, such as autoencoders, has also contributed to nonlinear unmixing. Shahid et al. [20] introduced an innovative cross-product layer to achieve reliable reconstruction of observed pixels in accordance with the BMMs. Su et al. [21] constructed a double AE structure to simultaneously estimate the linear and nonlinear components of the hierarchical BMMs under a multitask learning framework. In [22] the nonlinear mixing effects were modeled by two fully connected hidden layers with a 3D convolutional neural network (CNN) capturing the spatial-spectral information of HSI. To enhance physical interpretability, a fully convolutional deep AE network was combined with the Hapke model in [23], yielding better unmixing results for intimate mixtures.

It is noted that SV may create a substantial obstacle in the implementation of hyperspectral unmixing. SV is usually induced by factors such as environmental conditions (e.g., atmosphere, illumination, and topography) and materials' intrinsic properties (e.g., composition, morphology, and physicochemical attributes) [24,25]. Furthermore, the spectral signatures of land covers undergo alterations in response to various spatiotemporal distributions [26]. Due to the typical influence of SV, targets in shadow or surrounded by tall and bright objects may be unsuccessfully detected [27], and tree species could be wrongly classified because of the notable variability between canopies' spectra [28,29]. In unmixing, endmembers' spectral signatures belonging to the same class can vary significantly across different pixels. However, most existing nonlinear unmixing methods tend to ignore the issue of SV, despite its consideration in certain exploratory linear unmixing techniques. Particularly, the inherent endmember errors induced by SV could be considerably enlarged and propagated in nonlinear unmixing. It is meaningful to explore how SV affects the nonlinear unmixing process and construct an effective scheme to deal with the two issues simultaneously.

A common strategy for mitigating SV in linear unmixing involves representing an endmember class with multiple spectral signatures and identifying the optimal combination of endmembers to minimize reconstruction errors [30]. One typical method for this purpose is the widely recognized multiple endmember spectral mixture analysis (MESMA) [31]. However, this approach usually suffers from a high computational burden of combinatorial optimization. To this end, Heylen et al. [32] designed an alternating angle minimization method whose computational complexity increases linearly with the sizes of the endmember library. Moreover, some works leverage physically meaningful parameters to model SV within the framework of the LMM. Thouvenin et al. utilized a perturbation vector to model the wavelength-dependent variability of a certain endmember, resulting in the perturbed linear mixing model (PLMM) [33]. In contrast to the PLMM which fails to explain the principal scaling changes caused by various illuminated or topological conditions, an extended linear mixing model (ELMM) was proposed in [34] to multiply each reference endmember with a scaling factor to represent SV in every pixel. Based on the ELMM, some advanced methods with improvements such as reducing the reference endmember error [35] and retaining the spatial contextual information of abundances [36] have been

proposed recently. To better explain complex and nonuniform SV, an augmented linear mixing model (ALMM) [37] models the pixel-level scaling factors and other spectral variabilities simultaneously by using an endmember dictionary and a low-coherent additional dictionary, respectively, leading to more accurate abundance estimation. In addition, SV is often considered to be reduced in a specific feature subspace. Hong et al. [38] designed a subspace learning strategy to project observed hyperspectral data to a low-rank subspace where SV can be effectively removed. Fisher transformation [39] has been proved to be successful in alleviating the effects of SV [40–42]. Liu et al. [41] introduced an unmixing method based on orthogonal Fisher transformation and applied it to estimate fractional vegetation cover in semiarid areas. Xu et al. [42] combined linear discriminant analysis (LDA) with MESMA to obtain impervious surface fractions in urban areas. However, methods based on Fisher transformation require the construction of an endmember library before unmixing, and their performance, to a great extent, commonly depends on the completeness of the library.

Recently, DL-based methods have shown their potential in mitigating SV. Regarding endmembers as stochastic variables for SV representation, Zhao et al. [43] proposed a variational Bayesian method to learn the probability distribution of endmembers under a 3D CNN framework. Similarly, such probability distribution can also be derived from the deep generative models [44,45]. In contrast to these probability-based models, Hong et al. [46] devised a two-stream network using an additional branch to guide the network to yield physically meaningful unmixing results. Although the aforementioned methods have achieved some improvement in addressing SV, most of them (especially conventional model-based methods) are based on the LMM and fail to take the nonlinear mixing effects into account, which greatly limits their application in real-world scenarios.

Generally, in scenarios such as urban areas, multiple scattering between natural or artificial objects at different heights is significant, and SV widely exists in typical impervious surfaces [47]. SV occurs with nonlinear mixing effects, resulting in an increase in unmixing error. However, dealing with the two issues could be very challenging due to their coupled influence on unmixing. Eches et al. mathematically proved that the ELMM can be derived from the Hapke model [48], indicating that the physical properties (e.g., illumination and geometry) can not only affect the multiple scattering of intimate mixtures but also change the spectral shapes, and thus produce SV. SV causes deviations between true pixel-dependent endmembers and the traditional endmember set for the whole image. Such deviations could propagate endmember errors in the process of unmixing [26], especially by deteriorating the interpretability of high-order scattering terms in the NMMs.

To overcome the above issues, this paper develops a novel nonlinear unmixing method by incorporating Fisher transformation into the PPNM, and thus reducing the impact of SV. Firstly, an improved superpixel-based Fisher transformation with an abundance-driven coarse classification strategy is proposed. Its main target is to make the pixels' spectra which are classified to the same type of land cover similar, and enlarge the spectral difference between pixels belonging to different land covers. Particularly, since the transformation is built based on superpixels, an effective balance could be achieved between the pixels' spectral similarity and difference in local spatial homogenous regions. Secondly, with the PPNM being employed to explain the nonlinear mixing effects, we weighted two data reconstruction terms in both the original spectral space and the transformed subspace. In this sense, a constrained optimization problem for nonlinear unmixing is formulated by further using two specific regularizers of the projection matrix and the abundances. Finally, to improve the convergence and accuracy of calculation, a multi-swarm particle swarm optimization (PSO) algorithm [49] was exploited to estimate multiple unknown variables in the complex nonlinear unmixing problem, leading to a Fisher transformation-based unmixing algorithm via particle swarm optimization (FTUPSO).

The main contributions of our work are as follows:

- A superpixel-based Fisher transformation strategy is proposed to reduce the influence of spectral variability. In the transformed subspace, it enhances the similarity

between pixels corresponding to the same class and enlarges the difference between pixels belonging to different classes. Within-class and between-class scatter matrices are generated based on superpixels according to abundance-driven dynamic coarse classification, which can effectively reduce the impact of global misclassification and retain the similarity of pixels in local spatial homogenous regions.

- The improved Fisher transformation is combined with the PPNM to address the non-linear mixing effects and spectral variability simultaneously. Based on the PPNM, pixels are reconstructed in both the original spectral space and the weighted transformed subspace. With the incorporation of a projection matrix's regularization term and a TV-based regularizer of abundances, a novel unsupervised nonlinear unmixing problem is formulated, which can be regarded as a general framework for handling spectral variability in unmixing.

- Considering the complexity of the formulated unmixing problem, a dimensional division-based PSO is extended to solve the unknown unmixing variables. More reliable and accurate unmixing results can be produced by the proposed method.

The remainder of this paper is organized as follows. Section 2 introduces the LMM and PPNM, two classical approaches accounting for SV and the principle of Fisher transformation. Section 3 presents the details of the proposed method. Experimental results for both the synthetic and real datasets are provided in Section 4 and discussed in Section 5. Section 6 concludes the whole paper.

## 2. Related Works

### 2.1. LMM and PPNM

For a given HSI, the LMM interprets it as the linear combination of a set of endmember signatures and their fractional abundances, which can be formulated as follows:

$$X = EA + N, \tag{1}$$

where $X = [x_1, x_2, \ldots, x_N] \in \mathbb{R}^{M \times N}$ denotes the observed data matrix with $M$ spectral bands and $N$ pixels, $E = [e_1, e_2, \ldots, e_R] \in \mathbb{R}^{M \times R}$ denotes a matrix consisting of $R$ endmember vectors, $A = [a_1, a_2, \ldots, a_N] \in \mathbb{R}^{R \times N}$ represents the abundance matrix, and $N = [n_1, n_2, \ldots, n_N] \in \mathbb{R}^{M \times N}$ is the residual error. Abundances should usually satisfy two physical constraints, i.e., the abundance nonnegative constraint (ANC) and the abundance sum-to-one constraint (ASC):

$$A \geq 0, 1_R^T A = 1_N^T, \tag{2}$$

where $1_R \in \mathbb{R}^{R \times 1}$ and $1_N \in \mathbb{R}^{N \times 1}$ are two column vectors whose elements are all ones.

Different from the LMM's assumption, the PPNM further introduces second-order scattering terms to model the nonlinear contributions. The $j$th pixel based on the PPNM can be represented as:

$$x_j = Ea_j + b_j Ea_j \odot Ea_j + n_j, \ j = 1, 2, \ldots, N, \tag{3}$$

where $b_j$ is a bilinear parameter to adjust the contribution of nonlinearity and the operator $\odot$ denotes the Hadamard product. For the entire HSI, a compact matrix form of (3) can be written as:

$$X = EA + 1_M b^T \odot EA \odot EA + N, \tag{4}$$

where all the elements of the vector $1_M \in \mathbb{R}^{M \times 1}$ are one and $b = [b_1, b_2, \ldots, b_N]^T \in \mathbb{R}^{N \times 1}$.

### 2.2. ELMM and SULoRA

As a variant of the LMM, the ELMM exploits scaling factors to model SV, which allows endmembers to vary in every pixel:

$$X = E(S \odot A) + N. \tag{5}$$

In (5), the element of **S** in the *i*th row and *j*th column denotes the scaling factor of the *i*th endmember in the *j*th pixel. On the other hand, SULoRA aims to learn a low-rank subspace projection for reducing SV in the transformed space. The unmixing problem of SULoRA can be formulated using:

$$\min_{\mathbf{A},\mathbf{\Theta}} \frac{1}{2}\|\mathbf{\Theta}(\mathbf{X}-\mathbf{EA})\|_{\mathrm{F}}^2 + \frac{\alpha}{2}\|\mathbf{X}-\mathbf{\Theta}\mathbf{X}\|_{\mathrm{F}}^2 + \beta\|\mathbf{\Theta}\|_* + \gamma\|\mathbf{A}\|_{1,1}$$
$$\text{s.t. } \mathbf{A} \geq \mathbf{0},$$

(6)

where $\mathbf{\Theta} \in \mathbb{R}^{M \times M}$ is the transformation matrix, $\|\mathbf{A}\|_{1,1} = \sum_{i=1}^{R}\sum_{j=1}^{N}|\mathbf{A}_{i,j}|$, $\|\cdot\|_{\mathrm{F}}$ denotes the Frobenius norm, and $\|\cdot\|_*$ denotes the nuclear norm. $\alpha$, $\beta$, and $\gamma$ are small penalty coefficients.

### 2.3. Fisher Transformation

Fisher transformation projects training data into a subspace that minimizes the inner-class differences and maximizes the inter-class differences. Let $\mathbf{Y} = [\mathbf{y}_1, \mathbf{y}_2, \ldots, \mathbf{y}_N] \in \mathbb{R}^{D \times N}$ be $N$ training samples from $C$ classes, and $D$ is the input space's dimension. The Fisher transformation matrix can be obtained by solving the following optimization problem:

$$\min_{\mathbf{W}} \frac{trace(\mathbf{W}^{\mathrm{T}}\mathbf{S}_w\mathbf{W})}{trace(\mathbf{W}^{\mathrm{T}}\mathbf{S}_b\mathbf{W})},$$

(7)

where $\mathbf{W} \in \mathbb{R}^{D \times (C-1)}$ is the projection matrix, $\mathbf{S}_w$ and $\mathbf{S}_b$ are within-class and between-class scatter matrices. Specifically, $\mathbf{S}_w$ and $\mathbf{S}_b$ are derived using:

$$\begin{cases} \mathbf{S}_w = \sum\limits_{i=1}^{C} \sum\limits_{\mathbf{y}_j \in C_i} (\mathbf{y}_j - \mathbf{m}_i)(\mathbf{y}_j - \mathbf{m}_i)^{\mathrm{T}} \\ \mathbf{S}_b = \sum\limits_{i=1}^{C} N_i(\mathbf{m}_i - \mathbf{m})(\mathbf{m}_i - \mathbf{m})^{\mathrm{T}} \end{cases}.$$

(8)

In (8), **m** represents the mean vector of all training samples, $N_i$ is the number of samples belonging to the class $C_i$, and $\mathbf{m}_i$ denotes their mean vector. Usually, **W** is collected as $C-1$ eigenvectors corresponding to the first $C-1$ smallest eigenvalues of $\mathbf{S}_w^{-1}\mathbf{S}_b$.

## 3. Proposed Method

### 3.1. Superpixel-Based Fisher Transformation Using Abundance-Driven Dynamic Coarse Classification

Spectral variability can increase the difference between pixels of the same class and decrease the differences between those of different classes. Assuming all pixels are constituted by a common set of endmembers brings errors to the unmixing process, which can be significantly amplified in nonlinear unmixing problems containing multiple scatterings terms like (3). To deal with this issue, Fisher transformation is employed to improve traditional nonlinear unmixing and reduce the impact of spectral variability.

Notably, a predefined spectral library containing pure spectra of land covers is commonly required for Fisher transformation to learn the projection matrix **W** in (7), which is often inaccessible for unsupervised nonlinear unmixing. Therefore, we developed an abundance-driven coarse classification strategy to build a dynamically updated spectral library. During the unmixing, pixels with large abundances are recognized and classified into different classes, generating approximate $\mathbf{S}_w$ and $\mathbf{S}_b$ in (8) for Fisher transformation. Specifically, a pixel $\mathbf{x}_j$ can be coarsely classified into the category of its dominant endmember that has the maximum abundance:

$$c_j = \begin{cases} C_r, & \text{if } \max\{a_{i,j}\}_{i=1}^{R} \geq \varepsilon \text{ and } r = \underset{i}{\operatorname{argmax}}\{a_{i,j}\}_{i=1}^{R} \\ C_0, & \text{other} \end{cases},$$

(9)

where $a_{i,j}$ denotes the abundance of the endmember $\mathbf{e}_i$ in the pixel $\mathbf{x}_j$. $c_j$ denotes the class label of $\mathbf{x}_j$, and $C_r$ is the category of endmember $\mathbf{e}_r$ ($r = 1, 2, \ldots, R$), and $C_0$ is an

invalid category. Considering the influence of data's mixing degree, a threshold value $\varepsilon$ is introduced in (9) for the selection of approximate pure pixels. In terms of highly mixed data, it is possible that only a part of the pixels could be used for training **W**. As shown in Figure 1, the proposed method exploits dynamically updated abundances to determine clusters of land covers directly from HSIs for Fisher transformation, which could be consistent with the nature of hyperspectral data.

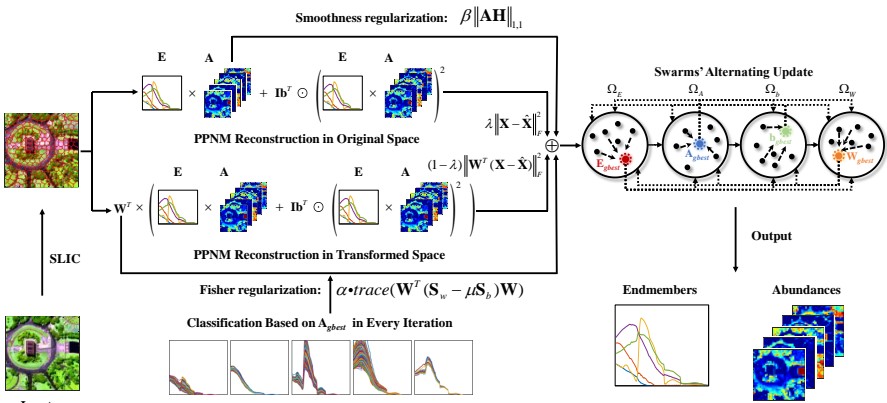

**Figure 1.** Schematic of the proposed FTUPSO method.

However, this strategy may also induce impractical estimation of neighboring objects' spatial distribution, especially for large heterogeneous scenarios. Two globally determined scatter matrices $\mathbf{S}_w$ and $\mathbf{S}_b$ can make the negative effects of misclassification serious and the optimization of **W** more sensitive to the value of $\varepsilon$. As a result, the difference in abundances between neighboring pixels within local spatial regions may increase excessively after the global Fisher transformation. It is in conflict with the common phenomenon that pixels within locally homogenous regions usually possess spectral similarity and similar abundances [50].

To overcome this limitation, the popular superpixel segmentation method SLIC [51] is exploited to improve the Fisher transformation. Let $\mathbb{P} = \{\mathbf{P}_1, \mathbf{P}_2, \ldots, \mathbf{P}_K\}$ denote $K$ superpixels, each of which contains a group of pixels in small, locally homogenous regions. Superpixel-based $\mathbf{S}_w$ and $\mathbf{S}_b$ can be further determined as follows:

$$\begin{cases} \mathbf{S}_w = \sum\limits_{k=1}^{K} \sum\limits_{r=1}^{R} \sum\limits_{\mathbf{x}_j \in C_r \text{ and } \mathbf{x}_j \in \mathbf{P}_k} (\mathbf{x}_j - \mathbf{m}_{r,k})(\mathbf{x}_j - \mathbf{m}_{r,k})^{\mathrm{T}} \\ \mathbf{S}_b = \sum\limits_{k=1}^{K} \sum\limits_{r=1}^{R} N_{r,k}(\mathbf{m}_{r,k} - \mathbf{m}_k)(\mathbf{m}_{r,k} - \mathbf{m}_k)^{\mathrm{T}} \end{cases}, \tag{10}$$

where $\mathbf{m}_k$ denotes the mean vector of pixels in $\mathbf{P}_k$, $\mathbf{m}_{r,k}$ is the mean vector of pixels belonging to $C_r$ in the superpixel $\mathbf{P}_k$ (pixels belonging to $C_0$ are excluded from both $\mathbf{m}_k$ and $\mathbf{m}_{r,k}$), and $N_{r,k}$ is the number of pixels belonging to $C_r$ in $\mathbf{P}_k$.

### 3.2. Nonlinear Unmixing Accounting for Spectral Variability

The PPNM is used to explain the nonlinear mixing effects. After superpixel-based Fisher transformation is incorporated into the process of unsupervised nonlinear unmixing, hyperspectral data are required to be approximately reconstructed in both the original spectral space and the transformed feature subspace. Thus, the final mathematical problem of nonlinear unmixing accounting for spectral variability can be formulated using:

$$\min_{\mathbf{W},\mathbf{E},\mathbf{A},\mathbf{b}} \lambda \left\| \mathbf{X} - \hat{\mathbf{X}} \right\|_{\mathrm{F}}^2 + (1-\lambda) \left\| \mathbf{W}^{\mathrm{T}}(\mathbf{X} - \hat{\mathbf{X}}) \right\|_{\mathrm{F}}^2 + \alpha \Phi(\mathbf{W}) + \beta \gamma(\mathbf{A})$$
$$s.t. \ \mathbf{A} \geq \mathbf{0}, \mathbf{1}_R^{\mathrm{T}}\mathbf{A} = \mathbf{1}_N^{\mathrm{T}}, \tag{11}$$

where $\overset{\wedge}{\mathbf{X}} = \mathbf{EA} + \mathbf{1}_M\mathbf{b}^{\mathrm{T}} \odot \mathbf{EA} \odot \mathbf{EA} \in \mathbb{R}^{M \times N}$ represents the PPNM-based reconstructed data. $\mathbf{W}$ is an $M \times M$ projection matrix in Fisher transformation. In (11), two data reconstruction terms are balanced by a nonnegative parameter $\lambda$. This structure requires pixels to be accurately approximated not only in the original spectral space but also in the transformed feature space. The first term is optimized to produce model-conforming endmembers and abundances. The second term further guides the searching direction into the solution space where SV can be reduced. In practice, according to the applications and observed hyperspectral data's property, a large $\lambda$ may be selected to make the proposed method work as a traditional PPNM-based unsupervised nonlinear unmixing method, while a small $\lambda$ makes a solution with prominent variance between different land covers' unmixing results which is preferred. A proper $\lambda$ can be set to mitigate the coupled effect of the nonlinearity and SV. Thus, the impact of SV on nonlinear unmixing can be reduced and satisfactory unmixing results can be obtained. $\Phi(\mathbf{W})$ and $\gamma(\mathbf{A})$ are two regularization terms for variables $\mathbf{W}$ and $\mathbf{A}$. $\alpha$ and $\beta$ are two small penalty parameters to adjust the influence of HSIs' structural features on unmixing. To make unmixing results have a more practical significance, $\Phi(\mathbf{W})$ and $\gamma(\mathbf{A})$ can be defined as follows:

(1)　Fisher Regularization term $\Phi(\mathbf{W})$: The transformation matrix $\mathbf{W}$ should satisfy the Fisher criterion, i.e.,

$$\Phi(\mathbf{W}) = \frac{trace(\mathbf{W}^{\mathrm{T}}\mathbf{S}_w\mathbf{W})}{trace(\mathbf{W}^{\mathrm{T}}\mathbf{S}_b\mathbf{W})}, \tag{12}$$

where scatter matrices $\mathbf{S}_w$ and $\mathbf{S}_b$ are obtained by (10). $\Phi(\mathbf{W})$ in (12) guarantees that the transformation can reduce the differences between pixels of the same class and the similarities between different materials. In contrast to traditional Fisher transformation-based methods calculating $\mathbf{W}$ in a preprocessing step, our proposed method trains it in an iterative unmixing process. Moreover, to facilitate the optimization of $\mathbf{W}$, (12) is rewritten as an approximate equivalent form [52,53]:

$$\Phi(\mathbf{W}) = trace(\mathbf{W}^{\mathrm{T}}(\mathbf{S}_w - \mu\mathbf{S}_b)\mathbf{W}) = \sum_{t=1}^{M} \mathbf{w}_t^{\mathrm{T}}(\mathbf{S}_w - \mu\mathbf{S}_b)\mathbf{w}_t, \tag{13}$$

where $\mathbf{w}_t$ denotes the $t$ th column vector of $\mathbf{W}$ and $\mu > 0$ is a small constant used to balance the contributions of $\mathbf{S}_w$ and $\mathbf{S}_b$.

(2)　TV Smoothness Regularization term $\gamma(\mathbf{A})$: In natural scenarios of hyperspectral data, the spatial distribution of land covers is often considered to be piecewise smooth, i.e., abundances of neighboring pixels in local homogeneous regions are similar. In this sense, the total variation (TV) regularization term [54] is further introduced to exploit the HSIs' spatial information and improve the smoothness of estimated abundances, which can be formulated using:

$$\gamma(\mathbf{A}) = \|\mathbf{AH}\|_{1,1}, \tag{14}$$

where $\mathbf{H} \in \mathbb{R}^{N \times 4N}$ is an operator utilized to calculate the differences between a given pixel and its four neighboring pixels (i.e., pixels on the top, bottom, left, and right). Notably, the settings of parameters $\lambda$, $\alpha$, and $\beta$ can depend on applications and the spatial-spectral structures of data for different observed scenarios.

### 3.3. Alternating Update of Unmixing Variables via Multi-Swarm PSO

There are four unknown variables, $\mathbf{W}$, $\mathbf{E}$, $\mathbf{A}$, and $\mathbf{b}$, that need to be estimated in the formulated unmixing problem (11). Commonly, traditional gradient-based alternating update methods are used, but they are easily trapped into the local optima, and produce unsatisfied unmixing results in solving nonconvex constrained optimization problems. However, the complexity and nonconvexity of (11) require a robust optimization method with global convergence to produce more accurate unmixing results. PSO has the superiority of easy

implementation and robustness to local optima in solving nonconvex constrained optimization problems, which has been investigated in the field of unmixing [55,56]. Considering the high-dimensional characteristics of hyperspectral unmixing, our previous work [49] developed a divide-and-conquer-based strategy to efficiently exchange particles' historical and global optimal information in different swarms according to the dimensions divided using the indices of pixels or bands. The dimensional division-based multi-swarm PSO can estimate particles' positions (i.e., the optimal solutions) more accurately in a finer search mode and work as a general framework to generate reliable results for different unmixing tasks [47,57]. Hence, the proposed method adopts this improved PSO framework for optimization.

Specifically, four swarms, denoted as $\Omega_E$, $\Omega_A$, $\Omega_b$ and $\Omega_W$, are constructed to optimize **E**, **A**, **b** and **W**, respectively. Each swarm has $Q$ particles. Assuming the other three variables are known, the optimization subproblems of these swarms are written as:

$$
\begin{cases}
\min_{\mathbf{E}} \lambda \left\| \mathbf{X} - \hat{\mathbf{X}} \right\|_F^2 + (1-\lambda)\left\| \mathbf{W}^T(\mathbf{X} - \hat{\mathbf{X}}) \right\|_F^2, & 0 \leq \mathbf{E} \leq 1 \\
\min_{\mathbf{A}} \lambda \left\| \mathbf{X} - \hat{\mathbf{X}} \right\|_F^2 + (1-\lambda)\left\| \mathbf{W}^T(\mathbf{X} - \hat{\mathbf{X}}) \right\|_F^2 + \beta\gamma(\mathbf{A}), & \mathbf{A} \geq \mathbf{0}, \mathbf{1}_R^T\mathbf{A} = \mathbf{1}_N^T \\
\min_{\mathbf{b}} \lambda \left\| \mathbf{X} - \hat{\mathbf{X}} \right\|_F^2 + (1-\lambda)\left\| \mathbf{W}^T(\mathbf{X} - \hat{\mathbf{X}}) \right\|_F^2, & \\
\min_{\mathbf{W}} (1-\lambda)\left\| \mathbf{W}^T(\mathbf{X} - \hat{\mathbf{X}}) \right\|_F^2 + \alpha\Phi(\mathbf{W}). &
\end{cases}
\tag{15}
$$

The fitness functions of four swarms are vectorized according to the matrices' rows or columns in (15). For example, the fitness vectors of swarm $\Omega_E$ can be generated by $M$ band-wise divisions, the fitness vectors of swarm $\Omega_W$ can be obtained by $M$ column-wise divisions, and the fitness vectors of swarms $\Omega_A$ and $\Omega_b$ can be obtained by $N$ pixel-wise divisions. In this sense, the fitness vectors of the four swarms are given in (16) and are detailed in (17)–(20):

$$
\begin{cases}
fitness(\mathbf{E}) = [f_1, f_2, \ldots, f_M], \\
fitness(\mathbf{A}) = [g_1, g_2, \ldots, g_N], \\
fitness(\mathbf{b}) = [h_1, h_2, \ldots, h_N], \\
fitness(\mathbf{W}) = [z_1, z_2, \ldots, z_M],
\end{cases}
\tag{16}
$$

and

$$
f_i = \lambda \sum_{j=1}^{N} (\mathbf{X} - \hat{\mathbf{X}})_{i,j}^2 + (1-\lambda) \sum_{j=1}^{N} (\mathbf{W}^T(\mathbf{X} - \hat{\mathbf{X}}))_{i,j}^2, i = 1, 2, \ldots, M,
\tag{17}
$$

$$
\begin{aligned}
g_j &= \lambda \sum_{i=1}^{M} (\mathbf{X} - \hat{\mathbf{X}})_{i,j}^2 + (1-\lambda) \sum_{i=1}^{M} (\mathbf{W}^T(\mathbf{X} - \hat{\mathbf{X}}))_{i,j}^2 \\
&\quad + \beta \sum_{i=1}^{R} \left( \left| (\mathbf{AH})_{i,j} \right| + \left| (\mathbf{AH})_{i,j+N} \right| + \left| (\mathbf{AH})_{i,j+2N} \right| \right. \\
&\quad \left. + \left| (\mathbf{AH})_{i,j+3N} \right| \right), j = 1, 2, \ldots, N,
\end{aligned}
\tag{18}
$$

$$
h_k = \lambda \sum_{i=1}^{M} (\mathbf{X} - \hat{\mathbf{X}})_{i,k}^2 + (1-\lambda) \sum_{i=1}^{M} (\mathbf{W}^T(\mathbf{X} - \hat{\mathbf{X}}))_{i,k}^2, k = 1, 2, \ldots, N,
\tag{19}
$$

$$
z_t = (1-\lambda) \sum_{j=1}^{M} (\mathbf{W}^T(\mathbf{X} - \hat{\mathbf{X}}))_{t,j}^2 + \alpha\mathbf{w}_t^T(\mathbf{S}_w - \mu\mathbf{S}_b)\mathbf{w}_t, t = 1, 2, \ldots, M.
\tag{20}
$$

Then, based on (16), particles compare their fitness vectors, and update their positions and velocities in corresponding dimensions using a velocity update equation [55] during the search for PSO. Notably, a particle's position represents a potential solution for **E**, **A**, **b** or **W** according to a specific swarm, $\Omega_E$, $\Omega_A$, $\Omega_b$ or $\Omega_W$. In the velocity update equation, each particle's current position and velocity, its best position in history, and the best position in its swarm

are used to calculate its new velocity and position following a simple physical momentum process. An inertia weight can be used to control the swarm's exploration and development, and two accelerating factors are often applied to balance the influence of cognitive and social learning of particles. Moreover, randomness is introduced to avoid the premature issue. In each iteration, each particle's best position in history and the best positions in the swarms could be updated. The detailed settings of PSO's hyperparameters, and the updated and boundary control rules of velocities and positions can be referred to [49].

In the proposed method, the positions of the particles in $\Omega_E$ are initialized by randomly selecting pixels from the HSI, and for the other three swarms, the initial positions are set as random values. The velocities of all particles are initialized as zeros. In order to accelerate convergence and better guide the search direction, four elite particles are added into the swarms. The positions of elite particles for $\Omega_E$ and $\Omega_A$ are provided by VCA [58] and FCLS [59], respectively. For $\Omega_b$, its elite particle's position is initialized by the least square solution of the first term in (11). The elite particle's position in $\Omega_W$ is set as the eigenvectors of $\mathbf{S}_w - \mu\mathbf{S}_b$ referring to [52]. The four initial global best positions $\mathbf{W}_{gbest}^{(0)}$, $\mathbf{E}_{gbest}^{(0)}$, $\mathbf{A}_{gbest}^{(0)}$, and $\mathbf{b}_{gbest}^{(0)}$ (superscript represents the current number of iterations) are determined by alternately updating the particles in the four swarms. As shown in Figure 1, for a given swarm, its fitness vectors are calculated by using the global best positions of the other three swarms, e.g., in every iteration, the global best position $\mathbf{W}_{gbest}^{(it)}$ of $\Omega_W$ is determined by using $\mathbf{E}_{gbest}^{(it)}$, $\mathbf{A}_{gbest}^{(it)}$, and $\mathbf{b}_{gbest}^{(it)}$ belonging to $\Omega_E$, $\Omega_A$ and, $\Omega_b$, respectively. The proposed FTUPSO can be briefly summarized in Algorithm 1.

---

**Algorithm 1: Fisher transformation-based unmixing algorithm via particle swarm optimization (FTUPSO)**

---

**Input:** Hyperspectral image $\mathbf{X} \in \mathbb{R}^{M \times N}$, parameters $\alpha$, $\beta$, and $\lambda$.
**Output:** Endmember matrix $\mathbf{E} \in \mathbb{R}^{M \times R}$, abundance matrix $\mathbf{A} \in \mathbb{R}^{R \times N}$, transformation matrix $\mathbf{W} \in \mathbb{R}^{M \times M}$, and bilinear parameter vector $\mathbf{b} \in \mathbb{R}^{N \times 1}$.
**Initialization:**
**1:** Perform the SLIC method on $\mathbf{X}$ to obtain $K$ superpixels.
**2:** Generate four swarms, $\Omega_W$, $\Omega_E$, $\Omega_A$, and $\Omega_b$, and initialize the particles' positions in the swarms and set the particle's velocities as zeros, and determine the global best positions, $\mathbf{W}_{gbest}^{(0)}$, $\mathbf{E}_{gbest}^{(0)}$, $\mathbf{A}_{gbest}^{(0)}$, and $\mathbf{b}_{gbest}^{(0)}$; $it = 0$.
**3: While** $it <$ maxiter, **perform** $it = it + 1$

    **3.1**: Calculate $fitness(\mathbf{E})$ for particles in $\Omega_E$ using $\mathbf{W}_{gbest}^{(it-1)}$, $\mathbf{A}_{gbest}^{(it-1)}$, and $\mathbf{b}_{gbest}^{(it-1)}$ from (17), then determine the global best position $\mathbf{E}_{gbest}^{(it)}$.

    **3.2:** Calculate $fitness(\mathbf{A})$ for particles in $\Omega_A$ using $\mathbf{W}_{gbest}^{(it-1)}$, $\mathbf{E}_{gbest}^{(it)}$, and $\mathbf{b}_{gbest}^{(it-1)}$ from (18), then determine the global best position $\mathbf{A}_{gbest}^{(it)}$.

    **3.3:** Calculate $fitness(\mathbf{b})$ for particles in $\Omega_b$ using $\mathbf{W}_{gbest}^{(it-1)}$, $\mathbf{E}_{gbest}^{(it)}$, and $\mathbf{A}_{gbest}^{(it)}$ from (19), then determine the global best position $\mathbf{b}_{gbest}^{(it)}$.

    **3.4:** Use $\mathbf{A}_{gbest}^{(it)}$ to classify $\mathbf{X}$ by (9), then calculate $\mathbf{S}_w$ and $\mathbf{S}_b$ using (10).

    **3.5:** Calculate $fitness(\mathbf{W})$ for particles in $\Omega_W$ using $\mathbf{E}_{gbest}^{(it)}$, $\mathbf{A}_{gbest}^{(it)}$, and $\mathbf{b}_{gbest}^{(it)}$ from (20), then determine the global best position $\mathbf{W}_{gbest}^{(it)}$.

    **3.6:** Update the velocities and positions for particles in four swarms using the historical and global best positions obtained in the previous iteration.

**End**

---

    **4:** Output $\mathbf{W}_{gbest}$, $\mathbf{E}_{gbest}$, $\mathbf{A}_{gbest}$, and $\mathbf{b}_{gbest}$.



## 4. Experiments and Results

In this section, experiments on synthetic datasets and two real hyperspectral images were conducted to validate the effectiveness of the proposed FTUPSO. Experimental results were compared with nine classical and state-of-the-art methods, including VCA, FCLS, NMF_QMV [60], SULoRA [38], ELMM, PGMSU [44], PPNM-GDA, MLMp [61], and Fan_NMF [62]. Regularization parameters in these methods were experimentally and empirically chosen to achieve the best performance according to the related references. VCA and FCLS were used for the initialization of unsupervised unmixing methods. All the experiments were run on MATLAB R2021b using a computer with a 3.80-GHz Intel® Core™ i7-10700K CPU and 64 GBs of memory.

### 4.1. Evaluation Metrics

In order to assess the performance of the compared methods, four quantitative metrics are utilized in this work. For synthetic datasets, spectral angle distance (SAD) in (21), and abundance overall root mean square error (aRMSE) in (22) were adopted to compare the accuracy of estimated endmembers and abundances. Due to the ground truth of endmembers and abundances being unavailable in the real hyperspectral data, a reconstruction error (RE) in (23) and a signal-to-reconstruction error (SRE) in (24) are employed to evaluate the fitting capability of the methods as auxiliary metrics like most published works. In (21) and (22), $\hat{\mathbf{e}}_r$ ($r = 1, 2, \ldots, R$) and $\hat{\mathbf{A}}$ denote the estimated endmembers and abundances, respectively.

$$\text{SAD} = \frac{1}{R}\sum_{r=1}^{R}\arccos\left(\frac{\mathbf{e}_r^{\mathsf{T}}\hat{\mathbf{e}}_r}{\|\mathbf{e}_r\|\|\hat{\mathbf{e}}_r\|}\right) \tag{21}$$

$$\text{aRMSE} = \sqrt{\frac{1}{RN}\sum_{j=1}^{N}\sum_{i=1}^{R}\left(\mathbf{A}_{i,j} - \hat{\mathbf{A}}_{i,j}\right)^2} \tag{22}$$

$$\text{RE} = \sqrt{\frac{1}{MN}\sum_{j=1}^{N}\sum_{i=1}^{M}\left(\mathbf{X}_{i,j} - \hat{\mathbf{X}}_{i,j}\right)^2} \tag{23}$$

$$\text{SRE} = 10\log_{10}\left(\frac{\|\mathbf{X}\|_{\mathrm{F}}^2}{\left\|\mathbf{X} - \hat{\mathbf{X}}\right\|_{\mathrm{F}}^2}\right) \tag{24}$$

### 4.2. Synthetic Data Experiments

To quantitatively compare the unmixing accuracy, synthetic datasets with different noise intensities, numbers of endmembers, and numbers of pixels were generated. Five endmembers were selected from the USGS spectral library (available at http://speclab.cr.usgs.gov/spectral-lib.html (accessed on 1 April 2023)) as the reference endmembers (shown in Figure 2) where each endmember is composed of 224 spectral bands covering wavelengths from 380 to 2500 nm. Then, the reference endmembers were multiplied with randomly generated scaling factors in the range [0.75, 1.25] to model SV for each pixel [37]. Abundances were sampled by the Gaussian random field (available at http://www.ehu.es/ccwintco/index.php/Hyperspectral_Imagery_Synthesis_tools_for_MATLAB (accessed on 1 April 2023)) method. The maximum abundance was set as 0.8 to simulate the highly mixed scenarios. Finally, pixels were mixed based on the PPNM with pixel-wise random bilinear parameters in the range [−0.3, 0.3], and additive Gaussian white noise was added. For a fair comparison, each method was run ten times independently, and the mean and standard deviation of the unmixing results were provided. The best experimental results are marked in bold.

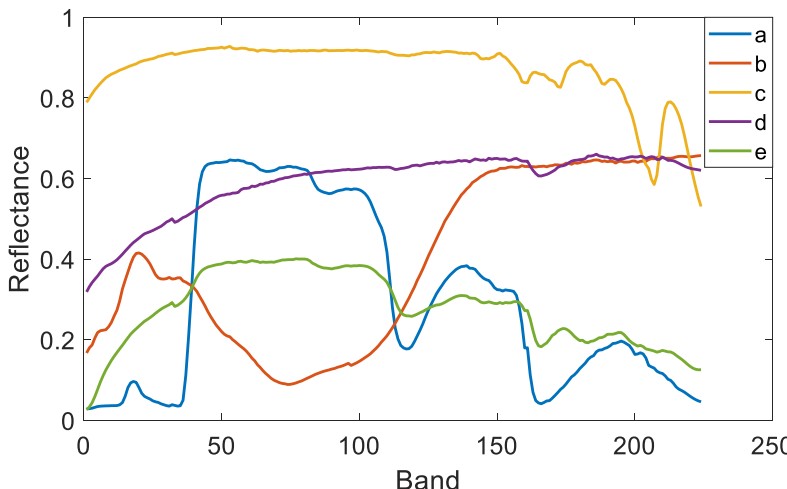

**Figure 2.** Reference endmembers selected from the USGS library. (**a**) Maple_Leaves DW92-1, (**b**) Olivine GDS70.a Fo89 165 um, (**c**) Calcite CO2004, (**d**) Quartz GDS74 Sand Ottawa, (**e**) Grass_dry.9+.1green AMX32.

(1)   Parameters' Settings

In the experiments, three parameters, $\lambda$, $\alpha$, and $\beta$ in (11), were determined to make the proposed method achieve the best performance. The accuracies of the endmember extraction and abundance estimation are considered simultaneously. In addition, for the SLIC method-based superpixel segmentation, the weight $w_s$ measuring spectral and spatial similarities was set as 0.5, and the average size of the superpixels was determined referring to [51]. A dataset comprising three endmembers and $32 \times 32$ pixels was first generated with the Signal to Noise Ratio ($SNR = 10 \log_{10}(E[\mathbf{x}^{\mathrm{T}}\mathbf{x}]/E[\boldsymbol{\varepsilon}^{\mathrm{T}}\boldsymbol{\varepsilon}])$) being 40 dB. Compared to $\alpha$ and $\beta$, $\lambda$ directly controls the minimization of reconstruction errors in the original space and the transformed feature space, which plays the most important role in the optimization problem of unmixing. Therefore, $\lambda$ was previously studied with $\alpha$ and $\beta$ being fixed. According to Figure 3, $\lambda$ could be set as 0.9. Then, based on the experimental results in Figures 4 and 5, $\alpha = 0.01$ and $\beta = 0.1$, the average size of the superpixels was 6.

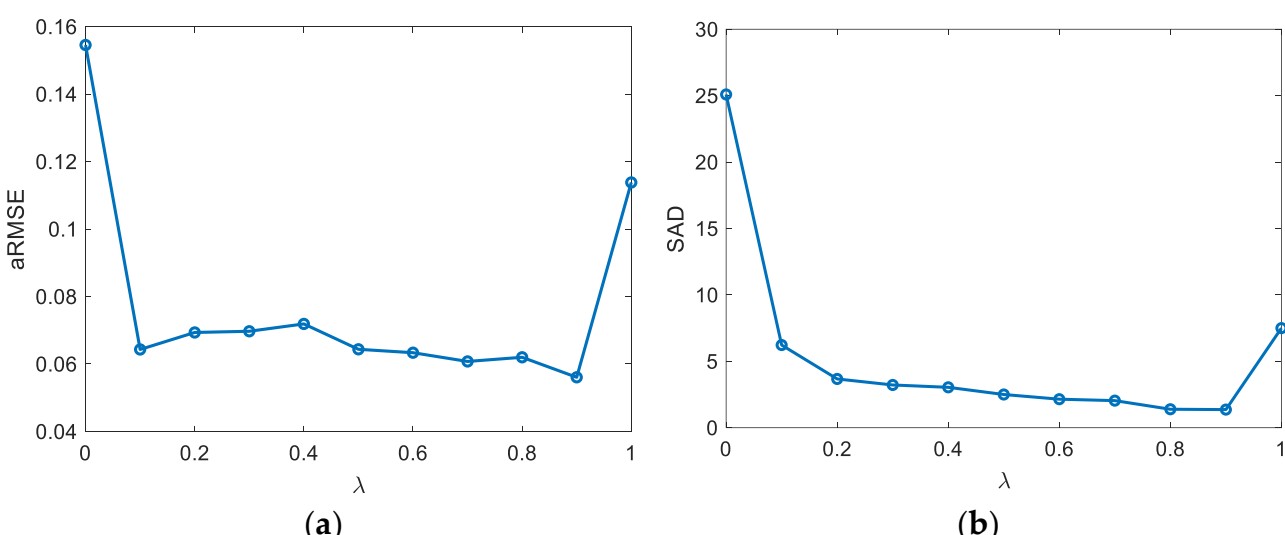

**Figure 3.** Influence of $\lambda$ on unmixing results. (**a**) aRMSE, (**b**) SAD.

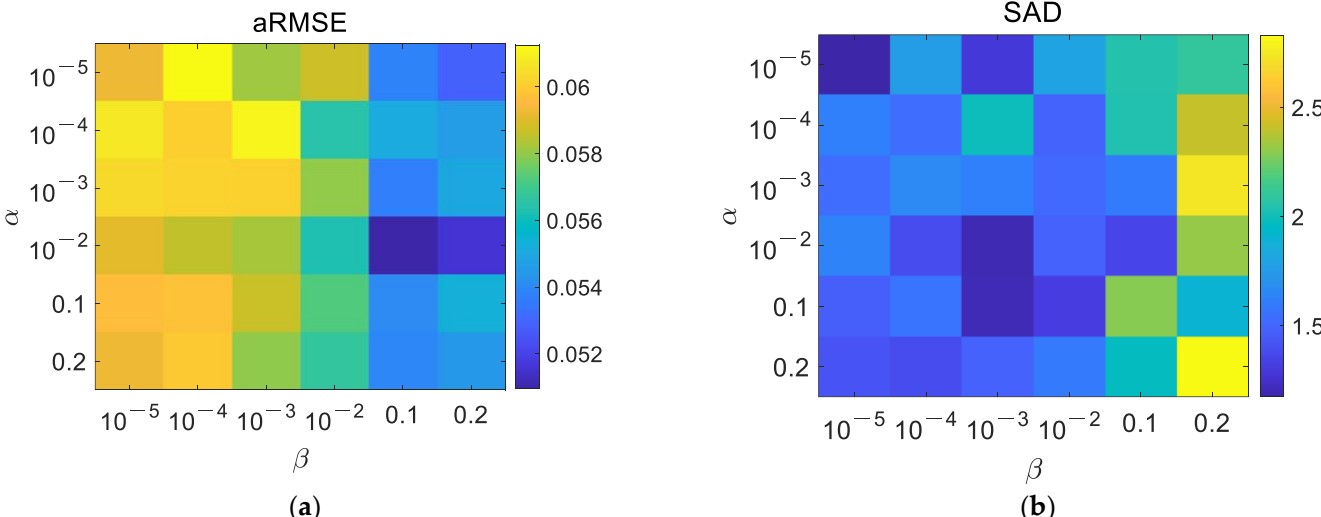

**Figure 4.** Influence of $\alpha$ and $\beta$ on unmixing results. (**a**) aRMSE, (**b**) SAD.

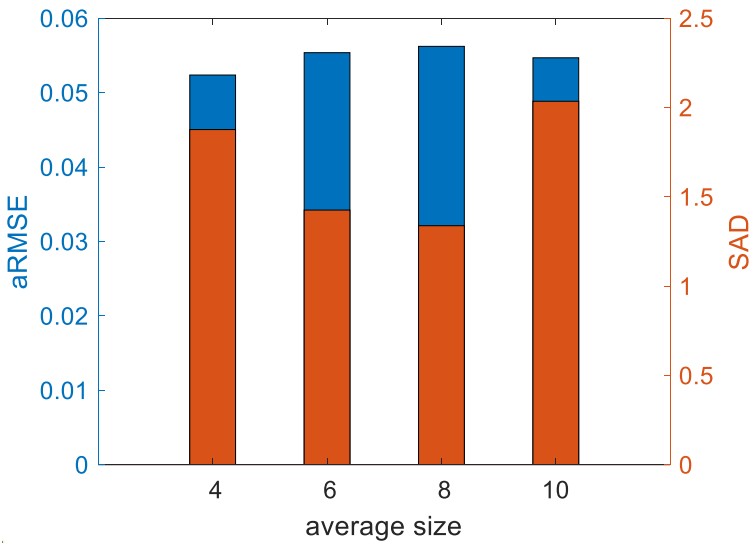

**Figure 5.** Influence of the average size of superpixels on aRMSE and SAD.

(2)    Ablation Experimental Analysis

To study the function of Fisher transformation, superpixel segmentation, TV regularization, and multi-swarm PSO on the proposed FTUPSO, ablation experiments were conducted for this section. First, a multi-swarm PSO method to optimize $\left\| \mathbf{X} - \hat{\mathbf{X}} \right\|_{\mathrm{F}}^{2}$ in (11) was implemented as the baseline. It actually solves a simple PPNM-based unsupervised nonlinear unmixing problem. Then, the other three ablation factors were added into the method in turn. The ablation analysis was performed on the same datasets as the parameter selection experiment to demonstrate the contribution of each of FTUPSO's parts.

In Table 1, the simplified baseline PSO method provides the poorest unmixing results. The adoption of a TV regularization term makes the unmixing accuracy outperform the baseline to a small degree due to the enhancement of the abundances' spatial smoothness. Moreover, as the Fisher transformation is further considered in the baseline framework, the errors of estimated endmembers and abundances are significantly reduced, indicating its effectiveness for addressing SV. Finally, it is clear that FTUPSO consisting of all four of the factors has the best unmixing performance. Superpixel improved Fisher transformation creates a good balance between the similarity and the difference of neighboring pixels.

**Table 1.** Ablation analysis on synthetic data.

| Metrics | aRMSE | SAD |
|---|---|---|
| PSO | $0.0724 \pm 0.0270$ | $3.6999 \pm 2.8524$ |
| PSO + TV | $0.0675 \pm 0.0244$ | $3.1646 \pm 2.4182$ |
| PSO + TV + Fisher | $0.0551 \pm 0.0160$ | $2.5460 \pm 2.0137$ |
| PSO + TV + Fisher + Superpixel | **$0.0529 \pm 0.0100$** | **$1.7551 \pm 0.6397$** |

The best experimental results are marked in bold.

(3)    Noise Robustness Analysis

In this experiment, datasets containing $32 \times 32$ pixels comprising three endmembers were generated to investigate the robustness of the compared methods to different noise intensities (i.e., SNR equals to 30 dB, 40 dB, and 50 dB). aRMSEs and SADs of estimated endmembers and abundances are listed in Table 2. Induced by the coupled effects of nonlinearity and SV, the unmixing results of traditional LMM-based methods (i.e., VCA, FCLS, and NMF_QMV) have larger unmixing errors than the other methods. ELMM, interpreting SV by scaling factors, improves the estimated abundances to some extent. However, it has similar SADs as VCA, implying its dependence on reference endmembers. SULoRA produces acceptable abundances because of its low-rank subspace transformation's robustness to SV. In contrast, PGMSU's aRMSEs are large and it fails to learn such a complex distribution of SV. It can be observed that most compared methods' unmixing results are slightly degraded at 50 dB compared to 40 dB, which may be induced by the randomness of generating data and noises. Compared to the linear unmixing methods, the nonlinear unmixing methods obtained more accurate unmixing results, but the ignorance of SV deteriorates their performance. MLMp cannot solve the endmembers correctly when the SNR is 30 dB. In conclusion, FTUPSO shows remarkable improvement for data with different SNRs. It may be inferred that the Fisher transformation is superior in reducing SV and makes FTUPSO robust to noises.

**Table 2.** Comparison of unmixing accuracies of all methods for synthetic data with different SNRs.

| SNR | 30 dB | | 40 dB | | 50 dB | |
|---|---|---|---|---|---|---|
| | aRMSE | SAD | aRMSE | SAD | aRMSE | SAD |
| VCA + FCLS | $0.1235 \pm 0.0277$ | $5.0554 \pm 1.0975$ | $0.1175 \pm 0.0250$ | $4.8533 \pm 0.2147$ | $0.1273 \pm 0.0294$ | $5.3393 \pm 0.8503$ |
| NMF_QMV | $0.1136 \pm 0.0104$ | $10.0529 \pm 2.1945$ | $0.1120 \pm 0.0105$ | $10.3264 \pm 1.5705$ | $0.1145 \pm 0.0087$ | $10.6626 \pm 2.1452$ |
| SULoRA | $0.0876 \pm 0.0200$ | - | $0.0853 \pm 0.0060$ | - | $0.0956 \pm 0.0220$ | - |
| ELMM | $0.0858 \pm 0.0231$ | $4.9727 \pm 1.0713$ | $0.0720 \pm 0.0046$ | $4.7724 \pm 0.2199$ | $0.0872 \pm 0.0230$ | $5.2443 \pm 0.8369$ |
| PGMSU | $0.1150 \pm 0.0132$ | $2.9392 \pm 0.4247$ | $0.1174 \pm 0.0100$ | $2.8098 \pm 0.6449$ | $0.1171 \pm 0.0127$ | $2.8701 \pm 0.6919$ |
| PPNM-GDA | $0.1087 \pm 0.0139$ | - | $0.1035 \pm 0.0082$ | - | $0.1085 \pm 0.0121$ | - |
| MLMp | $0.0945 \pm 0.0223$ | $10.9429 \pm 1.9940$ | $0.0860 \pm 0.0187$ | $6.1948 \pm 0.4053$ | $0.0967 \pm 0.0253$ | $6.1932 \pm 0.5542$ |
| Fan_NMF | $0.1090 \pm 0.0190$ | $3.3978 \pm 0.7028$ | $0.1085 \pm 0.0173$ | $3.4620 \pm 0.5106$ | $0.1139 \pm 0.0155$ | $3.5118 \pm 0.6926$ |
| FTUPSO | **$0.0646 \pm 0.0174$** | **$2.4315 \pm 1.0783$** | **$0.0553 \pm 0.0113$** | **$1.9133 \pm 0.6487$** | **$0.0668 \pm 0.0183$** | **$2.8610 \pm 1.5002$** |

The best experimental results are marked in bold.

(4)    Sensitivity Analysis to the Number of Endmembers

This section details experiments conducted to analyze the methods' sensitivity to the number of endmembers. Table 3 compares the aRMSEs and SADs obtained by the compared methods. Pixels consist of different numbers of endmembers, ranging from three to five. Each dataset has a total of $32 \times 32$ pixels and the SNR is 40 dB. In Table 3, the variation in the number of endmembers results in different trends in unmixing accuracies among the methods. NMF_QMV seems to be the most sensitive to the changes of the number of endmembers. In the cases where no more than four endmembers are considered, it nearly failed to extract accurate endmembers. However, FTUPSO always provides the best unmixing results for different datasets. It implies that, even for a complex high-

dimensional unmixing problem, the dimension-wise search of multi-swarm PSO makes the proposed method can obtain robust and accurate solutions.

**Table 3.** Comparison of unmixing accuracies of all methods for synthetic data with different numbers of endmembers.

| Endmembers' Numbers | 3 | | 4 | | 5 | |
|---|---|---|---|---|---|---|
| | aRMSE | SAD | aRMSE | SAD | aRMSE | SAD |
| VCA + FCLS | $0.1175 \pm 0.0250$ | $4.8533 \pm 0.2147$ | $0.1182 \pm 0.0127$ | $4.7575 \pm 0.3368$ | $0.1349 \pm 0.0251$ | $4.6193 \pm 0.4725$ |
| NMF_QMV | $0.1120 \pm 0.0105$ | $10.3264 \pm 1.5705$ | $0.2208 \pm 0.0305$ | $11.8767 \pm 8.7074$ | $0.1533 \pm 0.0173$ | $4.6940 \pm 2.0134$ |
| SULoRA | $0.0853 \pm 0.0060$ | - | $0.0924 \pm 0.0080$ | - | $0.1028 \pm 0.0219$ | - |
| ELMM | $0.0720 \pm 0.0046$ | $4.7724 \pm 0.2199$ | $0.1138 \pm 0.0159$ | $4.6558 \pm 0.3446$ | $0.1048 \pm 0.0130$ | $4.5240 \pm 0.4673$ |
| PGMSU | $0.1174 \pm 0.0100$ | $2.8098 \pm 0.6449$ | $0.1516 \pm 0.0180$ | $4.4599 \pm 0.8600$ | $0.1587 \pm 0.0115$ | $5.7992 \pm 0.7512$ |
| PPNM-GDA | $0.1035 \pm 0.0082$ | - | $0.1019 \pm 0.0071$ | - | $0.0897 \pm 0.0080$ | - |
| MLMp | $0.0860 \pm 0.0187$ | $6.1948 \pm 0.4053$ | $0.1025 \pm 0.0141$ | $6.5597 \pm 0.6135$ | $0.1238 \pm 0.0254$ | $4.9190 \pm 0.5242$ |
| Fan_NMF | $0.1085 \pm 0.0173$ | $3.4620 \pm 0.5106$ | $0.0948 \pm 0.0083$ | $3.3640 \pm 0.6885$ | $0.1134 \pm 0.0262$ | $3.6232 \pm 0.4499$ |
| FTUPSO | $\mathbf{0.0553 \pm 0.0113}$ | $\mathbf{1.9133 \pm 0.6487}$ | $\mathbf{0.0845 \pm 0.0102}$ | $\mathbf{3.2964 \pm 0.5120}$ | $\mathbf{0.0877 \pm 0.0108}$ | $\mathbf{2.9669 \pm 0.6588}$ |

The best experimental results are marked in bold.

(5)  Sensitivity Analysis to the Number of Pixels

In this experiment, the methods' sensitivity to the number of pixels is further studied in datasets generated using different numbers of pixels varying from $32 \times 32$ to $64 \times 64$. Three endmembers were used and SNR was set as 40 dB. Table 4 shows the quantitative evaluation of different methods. Consistent with the above experiments, NMF_QMV had large unmixing errors in most cases, and the other methods were outperformed by FTUPSO. Simulated data characterized by significant nonlinearity and SV may pose challenges for the compared methods and deteriorate their performance. Nevertheless, FTUPSO's estimated endmembers and abundances have the smallest aRMSEs and SADs. The data's size has no significant impact on the proposed method's unmixing accuracy.

**Table 4.** Comparison of unmixing accuracies of all methods for synthetic data with different numbers of pixels.

| Pixels' Numbers | $32 \times 32$ | | $48 \times 48$ | | $64 \times 64$ | |
|---|---|---|---|---|---|---|
| | aRMSE | SAD | aRMSE | SAD | aRMSE | SAD |
| VCA + FCLS | $0.1175 \pm 0.0250$ | $4.8533 \pm 0.2147$ | $0.1340 \pm 0.0214$ | $4.8910 \pm 0.2676$ | $0.1124 \pm 0.0126$ | $4.8456 \pm 0.4261$ |
| NMF_QMV | $0.1120 \pm 0.0105$ | $10.3264 \pm 1.5705$ | $0.1144 \pm 0.0095$ | $9.9620 \pm 2.6018$ | $0.1230 \pm 0.0149$ | $10.6720 \pm 2.2458$ |
| SULoRA | $0.0853 \pm 0.0060$ | - | $0.0930 \pm 0.0061$ | - | $0.0866 \pm 0.0063$ | - |
| ELMM | $0.0720 \pm 0.0046$ | $4.7724 \pm 0.2199$ | $0.0792 \pm 0.0096$ | $4.8209 \pm 0.2769$ | $0.0768 \pm 0.0074$ | $4.7691 \pm 0.4242$ |
| PGMSU | $0.1174 \pm 0.0100$ | $2.8098 \pm 0.6449$ | $0.1214 \pm 0.0124$ | $2.8836 \pm 0.7589$ | $0.1250 \pm 0.0182$ | $2.9379 \pm 0.6014$ |
| PPNM-GDA | $0.1035 \pm 0.0082$ | - | $0.1045 \pm 0.0082$ | - | $0.1006 \pm 0.0050$ | - |
| MLMp | $0.0860 \pm 0.0187$ | $6.1948 \pm 0.4053$ | $0.0991 \pm 0.0149$ | $5.9899 \pm 0.3767$ | $0.0836 \pm 0.0110$ | $6.3783 \pm 0.7071$ |
| Fan_NMF | $0.1085 \pm 0.0173$ | $3.4620 \pm 0.5106$ | $0.1238 \pm 0.0138$ | $3.7635 \pm 0.7783$ | $0.1034 \pm 0.0119$ | $3.6131 \pm 0.6002$ |
| FTUPSO | $\mathbf{0.0553 \pm 0.0113}$ | $\mathbf{1.9133 \pm 0.6487}$ | $\mathbf{0.0698 \pm 0.0111}$ | $\mathbf{2.8147 \pm 0.8162}$ | $\mathbf{0.0576 \pm 0.0079}$ | $\mathbf{2.1379 \pm 0.7995}$ |

The best experimental results are marked in bold.

(6)  Convergence Analysis and Time Cost Comparison

According to Algorithm 1, the time cost mainly results from the alternating update of four swarms, i.e., $\Omega_W$, $\Omega_E$, $\Omega_A$ and $\Omega_b$. Considering the computational burden for calculating fitness vectors, updating velocities and positions, and comparisons for determining the best particle, the total computational complexity of FTUPSO can be given as $\mathcal{O}(Q(MRN + MMN + RNN))$. The theoretical proof of PSO's convergence can be found in published works [55]. FTUPSO is a simple extension of traditional PSO and has similar convergence. In the experiments, the variations of FTUPSO's REs in both the original data space and the transformed feature subspace are depicted in Figure 6. It can be observed

that REs decrease monotonically as the number of iterations increases and they converge to small values. Hyperspectral data can be well fitted in the two spaces. Table 5 further lists the execution time of different methods for datasets with different numbers of endmembers and pixels. Due to the use of a swarm intelligence algorithm, which has a common drawback that accuracy gains may bring a high computational burden, the proposed method needs more time for unmixing compared to the other methods.

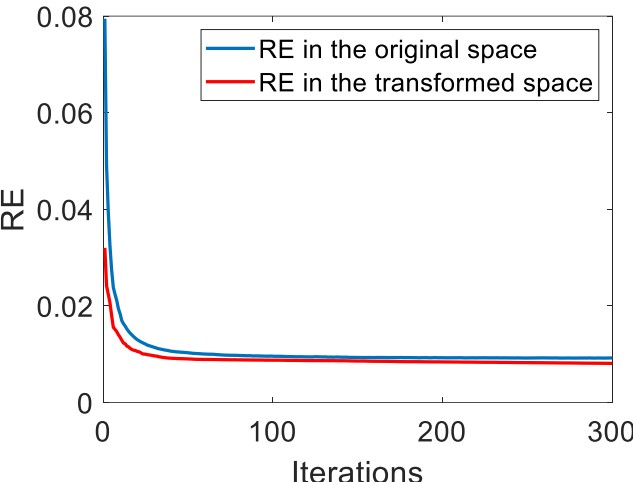

**Figure 6.** The change in the proposed method's REs in the original data space and the transformed feature space.

**Table 5.** Execution time (in seconds) of the methods for synthetic datasets generated using different numbers of endmembers and pixels.

| Execution Time | Number of Endmembers | | | Number of Pixels | | |
|---|---|---|---|---|---|---|
| | **3** | **4** | **5** | **32 × 32** | **48 × 48** | **64 × 64** |
| VCA + FCLS | **0.0434** | **0.0470** | **0.0696** | **0.0326** | **0.0650** | **0.1220** |
| NMF_QMV | 0.2431 | 0.2881 | 0.3406 | 0.2606 | 0.4014 | 0.6931 |
| SULoRA | 0.3645 | 0.3920 | 0.3853 | 0.3746 | 0.5447 | 0.7816 |
| ELMM | 8.0922 | 14.6351 | 18.7209 | 8.0323 | 21.8451 | 31.3933 |
| PGMSU | 15.7060 | 14.5231 | 14.6889 | 14.1952 | 15.7322 | 17.5270 |
| PPNM-GDA | 1.7015 | 4.9834 | 6.2781 | 1.6025 | 3.4623 | 7.7735 |
| MLMp | 2.1957 | 12.3033 | 8.6232 | 2.0675 | 4.5887 | 4.8350 |
| Fan_NMF | 3.4285 | 2.1612 | 2.9618 | 3.6760 | 6.1072 | 20.9294 |
| FTUPSO | 110.7060 | 109.8231 | 100.7271 | 111.5885 | 238.9956 | 416.8989 |

### 4.3. Real Hyperspectral Data Experiments

Two real hyperspectral remote sensing images were further applied to evaluate the performance of the proposed FTUPSO in practical scenarios. Due to the absence of ground truth, two quantitative metrics (i.e., RE and SRE) defined in (23) and (24) and qualitative experimental results including endmember curves and abundance maps were adopted to compare the overall unmixing accuracy of the methods. In addition, the execution time for each method was provided. In the following two experiments, considering the difference between the spatial structures of synthetic data and real hyperspectral data, we have set $\beta$ as 0.001 to alleviate the possible issue of over-smoothness. The other settings were the same as the synthetic data experiments.

(1) Washington DC Mall Dataset

The first hyperspectral image, captured by the Hyperspectral Digital Imagery Collection Experiment (HYDICE) sensor over Washington DC Mall, has 307 × 1280 pixels and a spatial resolution of 3 m. Spectral bands range from 0.4 to 2.5 μm. A subset of

165 bands was selected, excluding those impacted by noise (bands 1–15, 102–107, 137–152, and 203–210). A 100 × 100-pixel sub-image of this dataset was selected as the Region of Interest (RoI) for unmixing. Five categories of land covers exist: #1 Water, #2 Roof, #3 Tree, #4 Road, and #5 Grass. Figure 7 depicts the false color image and the superpixels produced by the SLIC method for this RoI.

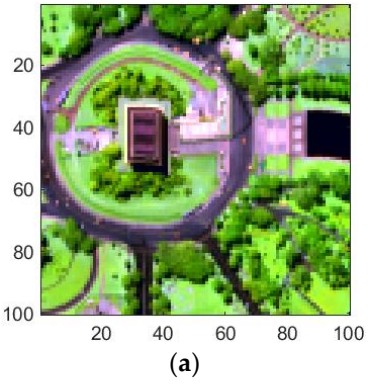
(**a**)

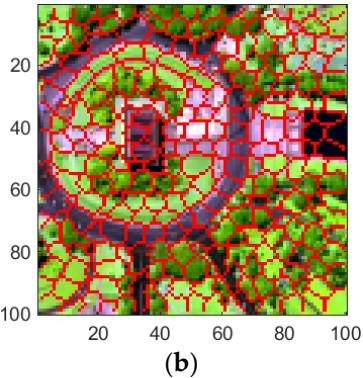
(**b**)

**Figure 7.** A 100 × 100-pixel RoI of Washington DC Mall data for experiments. The spatial resolution is 3 m. (**a**) False color image, (**b**) Result of superpixel segmentation.

Experimental results' SREs and REs for the Washington DC Mall data can be observed in Table 6. In terms of data reconstruction, the linear unmixing methods produced larger REs and smaller SREs than the nonlinear unmixing methods. ELMM can address the issue of SV and is considered to have the ability to explain nonlinear mixing effects to some extent [63], which makes it perform the best. Abundance maps and extracted endmembers' spectral curves are presented in Figures 8 and 9, respectively. The spatial distribution of water was recognized by all the methods, but was possibly overestimated by NMF_QMV and PGMSU in some pixels. SULoRA, incorrectly recognizing roads and water as the roof, produced the worst result. It is inferred that only enhancing the inner-class similarities but neglecting the inter-class differences may result in misidentification. Trees and grass were not clearly distinguished by the methods. However, FTUPSO, taking the advantages of Fisher transformation and superpixel segmentation, could estimate abundances that were more consistent with the real spatial distribution of most land covers. Moreover, due to the use of a TV regularization term, the abundance maps estimated by FTUPSO are smoother than the other methods' maps. In terms of endmember curves, NMF_QMV and PGMSU seem to overestimate the roof spectra's amplitude, and all the methods have similar results for the other land covers. The above comparison illustrates that FTUPSO can provide reasonable and accurate qualitative unmixing results due to its ability to reduce the impact of spectral variability in nonlinear unmixing.

**Table 6.** Comparison of the methods' unmixing performance for real hyperspectral data.

| Metrics | Washington | | | Cuprite | | |
|---|---|---|---|---|---|---|
| | **RE** | **SRE** | **Time** | **RE** | **SRE** | **Time** |
| VCA + FCLS | 0.0314 | 18.7345 | **0.7015** | 0.0082 | 33.0812 | 7.0573 |
| NMF_QMV | 0.0277 | 20.2062 | 1.6970 | 0.0440 | 19.1431 | 33.7064 |
| SULoRA | 0.0511 | 15.1141 | 0.7667 | 0.0981 | 13.3269 | **4.4248** |
| ELMM | **0.0047** | **35.5494** | 175.6399 | **0.0039** | 39.5010 | 770.7104 |
| PGMSU | 0.0096 | 29.4179 | 21.4256 | 0.0067 | 34.8311 | 60.6150 |
| PPNM-GDA | 0.0105 | 28.5890 | 63.4661 | 0.0059 | 35.8866 | 226.6430 |
| MLMp | 0.0121 | 27.4003 | 61.4246 | 0.0061 | 35.6612 | 53.2769 |
| Fan_NMF | 0.0090 | 29.9461 | 4.6028 | 0.0058 | 36.0014 | 84.9318 |
| FTUPSO | 0.0062 | 33.1701 | 762.0527 | **0.0039** | **39.5066** | 3484.3585 |

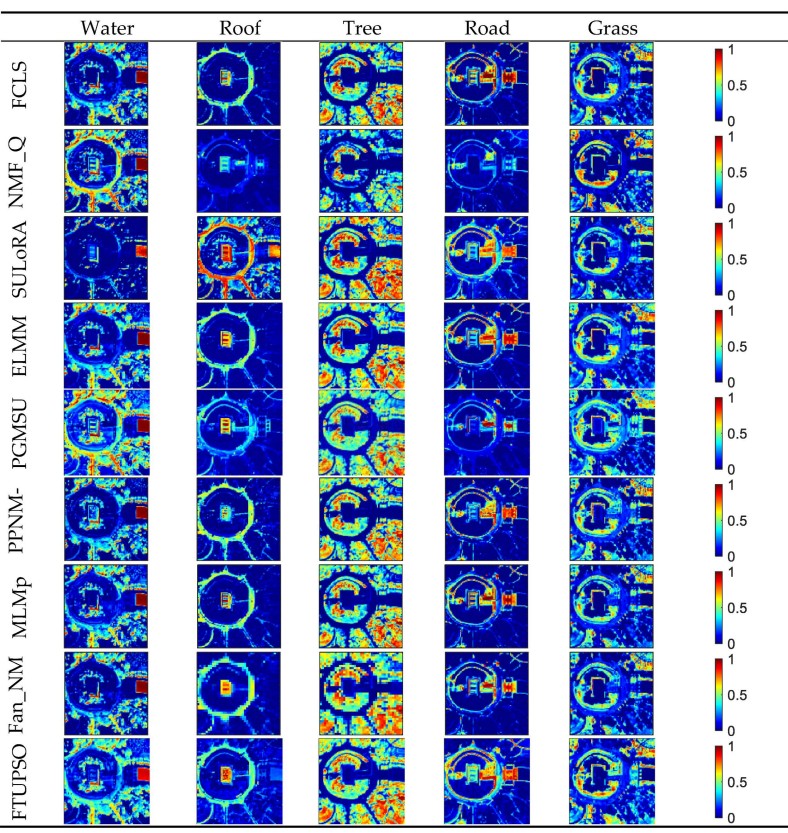

**Figure 8.** Abundance maps of Washington DC Mall data estimated by different methods.

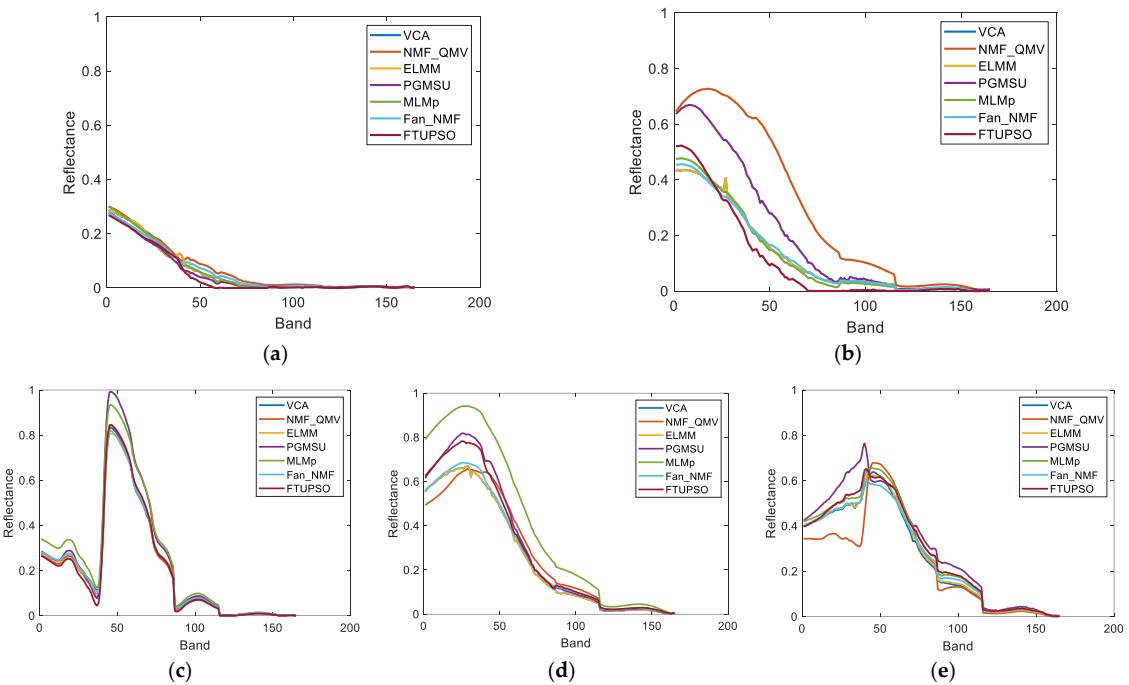

**Figure 9.** Endmembers of Washington DC Mall data extracted by different methods. (**a**) Water, (**b**) Roof, (**c**) Tree, (**d**) Road, (**e**) Grass.

(2)    Cuprite Dataset

The second real dataset was acquired by the Airborne Visible/Infrared Imaging Spectrometer (AVIRIS) over Cuprite, Nevada. A total of 224 spectral bands were sampled

ranging from 0.4 to 2.5 μm with a spectral resolution of 10 nm. Its ground sampling distance (GSD) was 20 m. This dataset's unique characteristics stem from the widespread, intimately mixed minerals, resulting in strong nonlinearity and spectral variability. It can be regarded as a good example for studying the impact of spectral variability on nonlinear unmixing. A region containing 200 × 200 pixels was selected for the experiments. Its false color image and superpixel segmentation image are displayed in Figure 10. After low SNR channels (1–2 and 221–224) and water absorption channels (104–113 and 148–167) were removed, 188 bands remained for unmixing. Twelve endmembers were considered in the experiments, i.e., #1 Alunite, #2 Sphene, #3 Kaolinite1, #4 Montmorillonite, #5 Kaolinite2, #6 Buddingtonite, #7 Pyrope, #8 Nontronite, #9 Muscovite, #10 Halloysite, #11 Chalcedony, and #12 Desert Varnish.

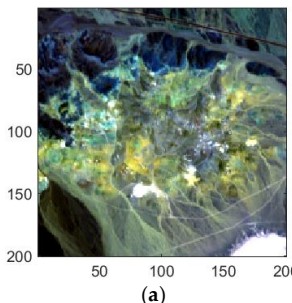 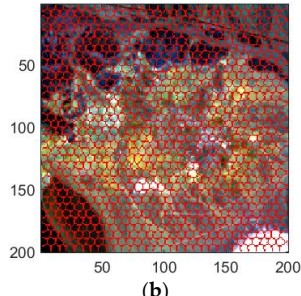

(**a**)  (**b**)

**Figure 10.** A 200 × 200-pixel RoI of Cuprite data for experiments. The spatial resolution is 20 m. (**a**) False color image, (**b**) Result of superpixel segmentation.

Table 6 compares the REs, SREs, and time costs of the methods. Due to the existence of strong nonlinear mixing effects, nonlinear unmixing methods provided better results (i.e., lower RE and higher SRE) than most linear unmixing methods except ELMM. FTUPSO performs better than the others, and the nonlinearity is well explained by its PPNM-based unmixing modules. Moreover, standard spectra of the USGS library were employed to quantitatively compare the extracted endmembers. Figure 11 provides the SADs between the reference endmembers (shown in Figure 12a) where the high spectral similarity can be observed, e.g., minerals such us Halloysite and Kaolinite have small SADs. Table 7 provides the SADs of the endmembers extracted using the compared methods. Affected by the intimate mixing effects and SV of different minerals, twelve endmembers with the smallest SADs were extracted by different methods. For example, FTUPSO provides the most accurate Alunite, Nontronite, and Muscovite, and Sphene and buddingtonite extracted by Fan_NMF have the smallest SADs.

**Table 7.** SADs of endmembers extracted by the methods for Cuprite data.

| Endmember | VCA | NMF_QMV | ELMM | PGMSU | MLMp | Fan_NMF | FTUPSO |
|---|---|---|---|---|---|---|---|
| Alunite | 5.0488 | 5.4230 | 5.0521 | 4.9842 | 5.0003 | 5.0614 | **3.8052** |
| Sphene | 3.4584 | 4.7743 | 3.4484 | 3.1775 | 3.5226 | **3.1221** | 4.4041 |
| Kaolinite1 | 10.3671 | 12.8995 | 10.3774 | **9.8591** | 10.5428 | 10.1228 | 12.6660 |
| Montmorillonite | **6.3181** | 7.0085 | 6.3290 | 9.0302 | 6.6698 | 6.8224 | 7.3845 |
| Kaolinite2 | 13.1916 | **10.3799** | 13.2048 | 15.8814 | 13.4972 | 12.8346 | 14.1055 |
| Buddingtonite | 6.2326 | 8.3854 | 6.2354 | 6.4710 | 6.2613 | **5.3374** | 6.5042 |
| Pyrope | 3.9504 | 7.3195 | 3.9129 | **2.6020** | 3.5621 | 3.4190 | 3.1795 |
| Nontronite | 5.1284 | 5.6919 | 5.1100 | 6.7004 | 5.0699 | 5.4248 | **4.9914** |
| Muscovite | 4.9336 | 5.7971 | 4.9339 | 7.7099 | 4.9387 | 4.9679 | **4.8148** |
| Halloysite | 19.2877 | 24.2131 | 19.2886 | **16.4654** | 19.2655 | 18.1978 | 19.8161 |
| Chalcedony | 7.7124 | **5.3233** | 7.7337 | 8.6724 | 7.9504 | 7.0210 | 8.5571 |
| Desert Varnish | 12.8298 | **11.5568** | 12.8295 | 13.0414 | 13.0119 | 13.6539 | 12.9964 |

Moreover, Figure 12 depicts the estimated endmember curves, and their corresponding abundance maps are shown in Figure 13. Similar unmixing results are obtained by most methods for minerals including Alunite, Montmorillonite, and Muscovite. SULoRA may overestimate Halloysite and fail to recognize Chalcedony and Desert Varnish because of prominent spectral variability between some minerals (i.e., high similarities of spectral shapes) according to Figure 11. However, the superpixel-based Fisher transformation makes FTUPSO distinguish similar spectra and produce reasonable unmixing results.

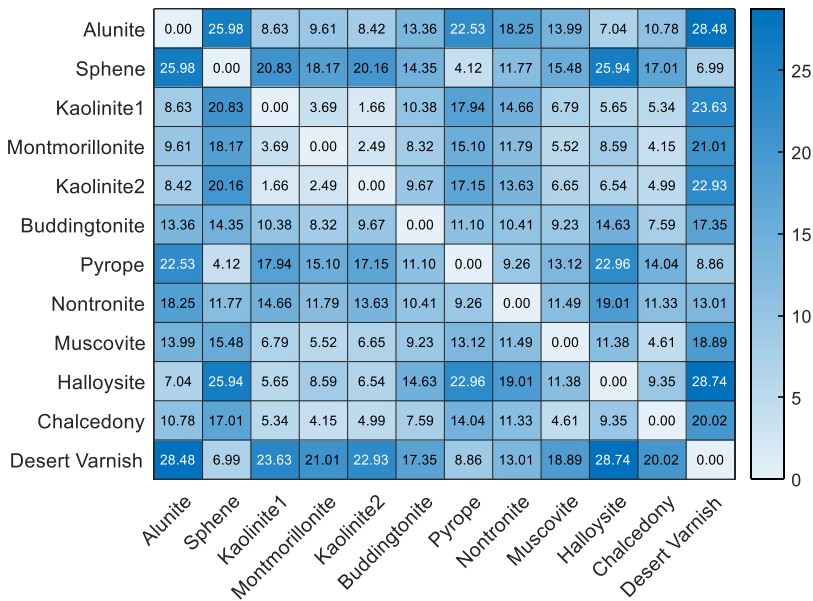

**Figure 11.** SADs between the reference endmembers in the USGS spectral library.

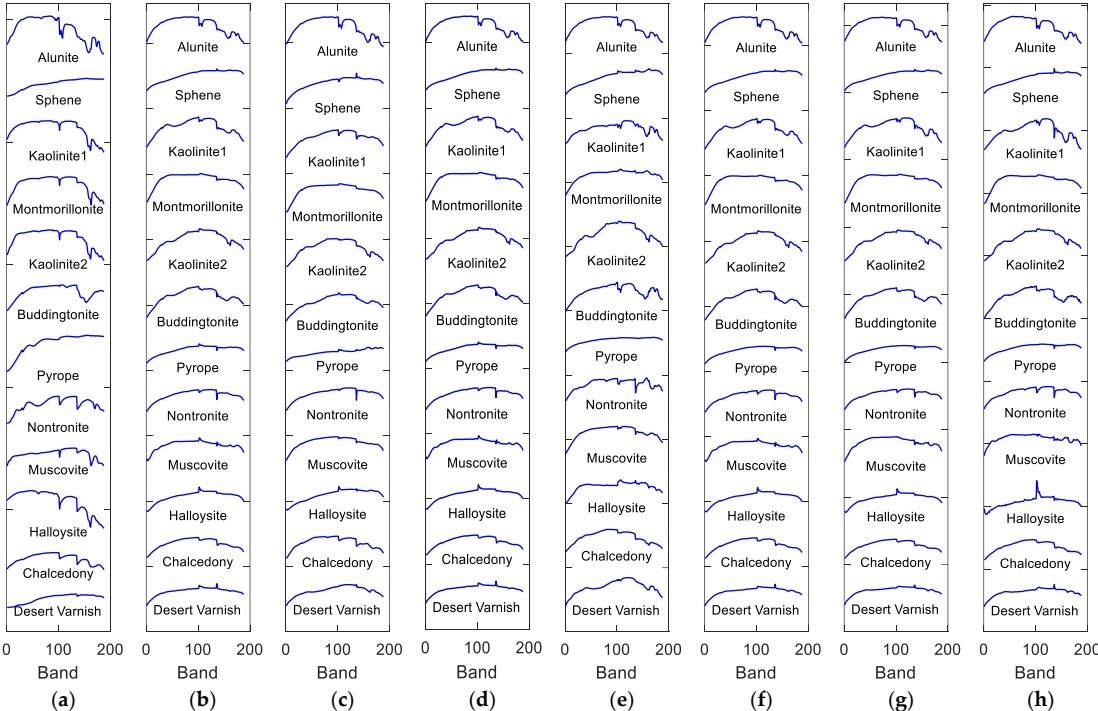

**Figure 12.** Endmembers of Cuprite data extracted by different methods. (**a**) Standard spectra, (**b**) VCA, (**c**) NMF_QMV, (**d**) ELMM, (**e**) PGMSU, (**f**) MLMp, (**g**) Fan_NMF, (**h**) FTUPSO.

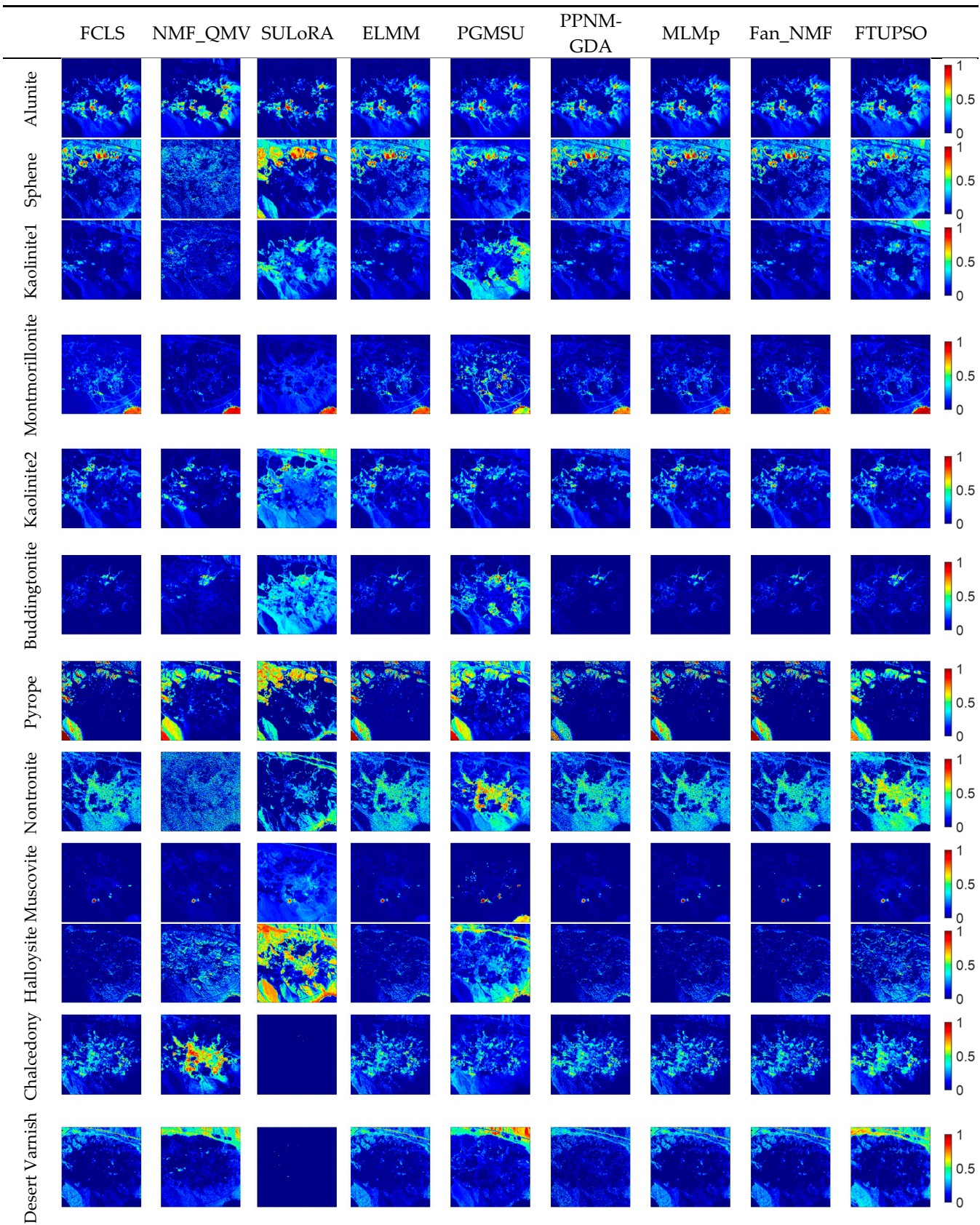

**Figure 13.** Abundance maps of Cuprite data estimated by different methods.

## 5. Discussion

To reduce the impact of spectral variability on nonlinear mixing, the proposed method incorporates the superpixel-based Fisher transformation into the PPNM-based unsupervised nonlinear unmixing framework. Accurate endmembers and abundances can be obtained by achieving a balance between the reconstruction error in the original data space and the reconstruction error in a transformed feature subspace with reduced SV. The quantitative and qualitative experimental results demonstrate that the proposed FTUPSO has remarkable unmixing performance in processing data with both nonlinearity and SV.

The results in Table 1 from the ablation experiments on synthetic data confirm the effectiveness of the four components of FTUPSO in improving unmixing accuracy. Quantitative comparison in Tables 2–4 prove that FTUPSO estimates more accurate endmembers and abundances than the other state-of-the-art methods. FTUPSO is not sensitive to the noise intensities, endmembers' numbers, and data sizes. Specifically, its robustness mainly results from two factors. First, the superpixel-based Fisher transformation balances reducing SV and improving similarities of different land covers in locally homogeneous regions. Second, the dimensional division-based multi-swarm PSO strategy further makes the proposed method yield better solutions than traditional gradient-based approaches.

In the real hyperspectral data experiments, FTUPSO is more effective in interpreting nonlinear mixing effects compared to methods based on the LMM. As shown in Figures 8 and 9, FTUPSO distinguishes land covers while mitigating the issues of overfitting or underestimation. Moreover, FTUPSO shows the ability to provide reasonable endmember curves and abundance maps. This is possibly because the superpixel-based Fisher transformation can not only enhance the spectral similarity in local homogeneous regions but can also enlarge the differences between the pixels belonging to different classes, which makes FTUPSO produce more accurate results. However, FTUPSO did not perform the best in the data fitting for the Washington DC Mall data, which may have been induced by the tradeoff between its two data reconstruction terms. Notably, in real situations, a smaller reconstruction error does not always mean a better estimation of endmembers and abundances. Table 7 and Figures 12 and 13 further verify that FTUPSO has competitive unmixing performances. In the experiments for Cuprite data, some regions of a mineral's abundance map can have pixels with large abundances, and pure pixels may exist. Although Halloysite estimated by SULoRA has large abundances in large parts of regions, it is not in line with references according to the published results [49]. In Figure 11, we can observe that Halloysite and minerals such as Alunite and Kaolinite have small SADs, implying that they have high spectral correlation. Other minerals also have similar characteristics. The abundance maps in Figure 13 indicate that they also have close spatial distribution. In contrast to Alunite and Kaolinite which have high abundances in a part of pixels, Halloysite seems to be highly mixed with some minerals. Therefore, considering the above factors, most unmixing methods cannot distinguish the spectral curve of Halloysite accurately because the extraction of Halloysite's spectra may be interfered with by other minerals.

The determination of the appropriate number of endmembers is important for unmixing [64]. In the synthetic data experiments, as the number of endmembers increases, most compared methods' unmixing accuracies decrease, but FTUPSO performs the best. In the experiments for the HYDICE data containing five endmembers, almost all the compared methods can identify every kind of land cover because of the prominent difference between the endmembers and the presence of pure pixels. However, in terms of the Cuprite data experiments, twelve endmembers should be considered; pure pixels are absent for several minerals, and the collinearity between endmembers increases significantly. In this situation, some unmixing methods such as SULoRA failed to estimate accurate endmembers and abundance maps. On the other hand, it is known that the HYDICE dataset has often been used for evaluating unmixing methods in many published works [10,49,56]. Although only five land covers (i.e., water, roof, tree, road, and grass) are taken into account, this image is very valuable for evaluating unmixing methods. The reason could be that the coarsely defined five endmembers can conveniently introduce the issue of SV into the

unmixing process for this image. For example, following such a definition of endmembers, typical land covers such as roofs, trees, and grasses can have strong SV in this observed scene. Their spectra vary significantly in different pixels. Therefore, it is meaningful to employ this image to validate the ability of the proposed method to reduce the impact of SV on unmixing.

Compared to the AVIRIS and HYDICE data used in this paper, space-borne data such as EO-1 Hyperion images [65] have lower spatial resolution and larger observation areas, and the cross-track illumination that includes the nonuniform illumination in the cross-track direction [66] and the deviation of the central wavelength position [67] becomes strong. Due to the impact of cross-track illumination, the inherent observation errors of pixels increase [67,68], indicating that further studies could be conducted to validate the proposed method's performance in addressing the issues caused by the cross-track illumination using Hyperion data.

## 6. Conclusions

This paper presents a novel unsupervised unmixing method addressing both nonlinear mixing effects and spectral variability. The proposed FTUPSO improves the traditional Fisher transformation by dynamic coarse classification and superpixel segmentation to reduce the impact of SV. Then, based on the PPNM, hyperspectral data are reconstructed in both the original and transformed spaces. With a TV regularizer being added to improve the smoothness of estimated abundances, the weighted minimization of two data reconstruction terms is achieved in an extended multi-swarm PSO algorithm to accelerate convergence and search for accurate unmixing results. Experimental results on several synthetic data and two real hyperspectral images demonstrate the superiority of FTUPSO in unsupervised unmixing compared to traditional and state-of-the-art methods.

However, FTUPSO may suffer from some limitations. For example, more advanced mechanisms can be developed to address extremely highly mixed data to avoid the possible wrong classification. Moreover, since only the scaling factor is considered in this work, other types of SV (e.g., perturbations) can affect the unmixing process and further study is required to overcome the limitations. In addition, in our next work, we will exploit FTUPSO to solve practical application problems such as urban impervious surface detection. Since the mechanism of FTUPSO is extremely time consuming, we will also accelerate it by using high-performance parallelization techniques.

**Author Contributions:** Conceptualization, Z.Y. and B.Y.; methodology, Z.Y. and B.Y.; software, Z.Y.; validation, Z.Y.; formal analysis, Z.Y. and B.Y.; data curation, Z.Y.; writing—original draft preparation, Z.Y.; writing—review and editing, B.Y.; visualization, Z.Y. and B.Y.; supervision, B.Y. All authors have read and agreed to the published version of the manuscript.

**Funding:** This work was supported by the National Natural Science Foundation of China under Grant No. 62001098, and by the Natural Science Foundation of Shanghai under Grant No. 23ZR1402400.

**Data Availability Statement:** The Cuprite and Washington DC Mall hyperspectral image datasets used in this study are freely available at http://rslab.ut.ac.ir/data (accessed on 1 June 2023).

**Conflicts of Interest:** The authors declare no conflict of interest.

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
