# Peer review of "Unsupervised Nonlinear Hyperspectral Unmixing with Reduced Spectral Variability via Superpixel-Based Fisher Transformation"

_remotesensing, doi:10.3390/rs15205028_

Round 1

Reviewer 2 Report

This paper a novel unsupervised nonlinear unmixing method accounting for SV,which has many innovation, the experiments and analysis are fruitful. There are some comments:

1) the authors in order to combine the consideration of nonlinearity and SV, that just use the add strategy maybe not enough, and how the proposed method simultaneous reduce the influence of nonlinearity and SV.

2) the PSO is used to produce accurate unmixing results, which should give more motivations, and how to design the structure, does the object optimized one by one, or it is a multi-object problem, and the convergence, et al.

3) as the results part, the evaluation metrics are not consistent in all datasets, which index can demonstrate the paper’s contribution.

4) the paper should give the whole structure and format the writing, polish the English.

Need polish.

Reviewer 3 Report

The necessary revisions and comments for authors;

I have re-read and re-evaluated the paper entitled as “Unsupervised Nonlinear Hyperspectral Unmixing with Reduced Spectral Variability via Superpixel-based Fisher Transformation”. 

There are some major to minor concerns and comments about the submitted article which listed as follows;

1)      It is better to give clearly what is the former one in line 49? Because, the paragraph started that “to explain the former,…”. Please write the name of former one. It is new paragraph which is newly started. What is the former one of what is the later one? It’s gone in previous paragraph.

2)      Figure 7 and 10 are too small to see and they must have scale bars to imagine how big the area. Please put the scale bars for both figures.

3)      Figure 8 and 12 show the abundance maps of different materials. However, which color shows us most abundant material? Is it red of blue? It is better to give rainbow color legend to show the most and least abundant in the images.

4)      This study is very well presented, I love it. However, some parts must be improved and some parts should be explained in the text.  In Cuprite image, halloysite mineral can be clearly seen or most abundant in SULoRA method. There are also similar scene for other minerals and methods. So why? Could you explain in discussion section about this manner?

5)      In figure 11, endmembers of Cuprite data from different methods were given. It is better to give original spectral signature data of the minerals from spectral libraries such as USGS, JPL or John Hopkins spectral libraries. Then we can understand which one is close to original spectral signature.

6)      The subject is so interesting but the paper have high similarity ratio (29%). Please check it and re-write some parts by your sentences, it can be defined plagiarism.

7)      Both HYDICE and AVISIS are airborne hyperspectral sensors. Therefore, cross-track illumination (or smile effect) is exist but it is minimum when comparing to space-borne hyperspectral sensors such as EO-1 Hyperion. The most of the mineral mapping studies used space-borne Hyperion sensor due to its availability all around the world. Please consider this subject in discussion section. Cross-track illumination of hyperspectral sensors also affects the spectral mixing of the observed materials. You can find details on following resources: Evaluation of cross-track illumination in EO-1 Hyperion imagery for lithological mapping (International Journal of Remote Sensing, 2011), Preprocessing EO1 Hyperion hyperspectral data to support the application of agricultural indexes (IEEE Transactions on Geoscience and Remote Sensing, 2003), Vicarious radiometric calibration of EO-1 sensors by reference to high-reflectance ground targets (IEEE Transactions on Geoscience and Remote Sensing, 2003), Cross-Track Illumination Correction for Hyperspectral Pushbroom Sensor Images Using Low-Rank and Sparse Representations (IEEE Transactions on Geoscience and Remote Sensing, 2023) etc.

8)      HYDICE image (Washington) was used for classification of water, roof, tree, road and grass. It has only 5 classes. AVIRIS image (Cuprite) had 12 classes. Therefore, number of classes and also spectral mixture types were different from each other. This case can be discussed in discussion section. In addition, the classification of only 5 distinct classes using hyperspectral data is logical or not? Hyperspectral remote sensing is generally suitable for the classification of lithological units and agricultural crops. Distinguishing 5 indistinguishable classes with hyperspectral datasets is not very economical. This can also be mentioned in the discussion section.

Round 2

Reviewer 2 Report

The authors have addressed my concerts.

The authors can check carefully, so as to pass the edit successful.

Reviewer 3 Report

All my comments and concerns were discussed in the text and they were clearly explained. THnak for your efford.